METHODS AND RESOURCES

# Assessment of single-vessel cerebral blood velocity by phase contrast fMRI

Xuming Chen[1,2⊛], Yuanyuan Jiang[3⊛], Sangcheon Choi[1,4], Rolf Pohmann[1], Klaus Scheffler[1,5], David Kleinfeld[6,7], Xin Yu[3]*

**1** High-Field Magnetic Resonance, Max Planck Institute for Biological Cybernetics, Tübingen, Germany, **2** Department of Neurology, Wuhan University, Renmin Hospital, Wuhan, China, **3** Athinoula A. Martinos Center for Biomedical Imaging, Massachusetts General Hospital and Harvard Medical School, Charlestown, Massachusetts, United States of America, **4** Graduate Training Centre of Neuroscience, International Max Planck Research School, University of Tübingen, Tübingen, Germany, **5** Department for Biomedical Magnetic Resonance, University of Tübingen, Tübingen, Germany, **6** Department of Physics, University of California at San Diego, La Jolla, California, United States of America, **7** Section of Neurobiology, University of California at San Diego, La Jolla, California, United States of America

⊛ These authors contributed equally to this work.
* xyu9@mgh.harvard.edu

**Data Availability Statement:** All relevant data are within the paper and its Supporting Information files.

**Funding:** X.Y. and Y.J. were supported by NIH RF1NS113278, R01NS120594, D.K. was

## Abstract

Current approaches to high-field functional MRI (fMRI) provide 2 means to map hemodynamics at the level of single vessels in the brain. One is through changes in deoxyhemoglobin in venules, i.e., blood oxygenation level–dependent (BOLD) fMRI, while the second is through changes in arteriole diameter, i.e., cerebral blood volume (CBV) fMRI. Here, we introduce cerebral blood flow–related velocity-based fMRI, denoted CBFv-fMRI, which uses high-resolution phase contrast (PC) MRI to form velocity measurements of flow. We use CBFv-fMRI in measure changes in blood velocity in single penetrating microvessels across rat parietal cortex. In contrast to the venule-dominated BOLD and arteriole-dominated CBV fMRI signals, CBFv-fMRI is comparable from both arterioles and venules. A single fMRI platform is used to map changes in blood $pO_2$ (BOLD), volume (CBV), and velocity (CBFv). This combined high-resolution single-vessel fMRI mapping scheme enables vessel-specific hemodynamic mapping in animal models of normal and diseased states and further has translational potential to map vascular dementia in diseased or injured human brains with ultra–high-field fMRI.

## Introduction

Cerebral blood flow (CBF) is a key hemodynamic readout coupled to neuronal dynamics and viability in normal and diseased brain states [1]. Changes in CBF may be monitored directly within individual blood vessels through the use of optical-based particle tracking techniques [2]. A variety of imaging methods have been developed to measure CBF across multiple spatial scales, from capillary beds up through brain-wide vascular networks in animal brains. These include in vivo multiphoton microscopy [3], optical coherence tomography [4], optoacoustic imaging [5], and laser doppler and speckle imaging [6,7]. In particular, doppler based

supported by NIH R35NS097265 and
R01MH111438, X.C. was supported by the student
scholarship from the Chinese Scholarship Council
and Deutsche Forschungsgemeinschaft (DFG,
Germany Research Foundation) grant YU 215/3-1,
K.S. and R.P. were supported by DFG SCHE 658/
15, SCHE 658/12. S.C. was supported by
Bundesministerium fuer bildung und forschung
(BMBF, Federal Ministry of Education and
Research) grant 01GQ1702. The funders had no
role in study design, data collection and analysis,
decision to publish, or preparation of the
manuscript.

**Competing interests:** The authors have declared
that no competing interests exist.

**Abbreviations:** AD, Alzheimer disease; AFNI,
Analysis of Functional NeuroImages; ASL, arterial
spin labeling; A–V, arteriole–venule; BOLD, blood
oxygenation level–dependent; bSSFP, balanced
steady-state free precession; CBF, cerebral blood
flow; CBFv, cerebral blood flow–related velocity;
CBV, cerebral blood volume; CNR, contrast-to-
noise ratio; fMRI, functional MRI; MGE, multi-
gradient echo; PC, phase contrast; RF, radio
frequency; SNR, signal-to-noise ratio; TE, echo
time; TR, repetition time; VASO, vascular-space-
occupancy; Venc, velocity encoding.

functional ultrasound imaging provides a unique advantage to detect the CBF in the brain with a high spatiotemporal resolution, which can be readily applied for awake animal imaging [8–10]. However, the spectrum-specific signal transmission in ultrasound methods cannot effectively pass the skull of animals without significant degradation of the signal-to-noise ratio (SNR). Typically, a craniotomy or a procedure to thin the skull is needed to detect the hemodynamic signal [2]. While current techniques support transcranial imaging into the superficial layers of the cortex [11–13], only functional MRI (fMRI) provides a noninvasive approach for measuring hemodynamic signals throughout the brain.

Changes in CBF may be detected by fMRI that is based on arterial spin labeling (ASL), in which water protons in a major upstream vessel are spin-polarized with an additional radio frequency (RF) B1 field [14–16]. The ASL-based CBF fMRI technique detects local changes in the flow of blood through brain tissue [17]. Two other fMRI-based techniques provide indirect information about changes in CBF. Blood oxygenation level–dependent (BOLD) fMRI is used to determine changes in the ratio of deoxy to oxyhemoglobin in the blood and is an indirect measure of changes in brain metabolism [14,18,19]. Cerebral blood volume (CBV) fMRI is used to measure changes in blood volume, i.e., essentially changes the diameter of arterioles, based on the use of exogenous contrast agents, e.g. monocrystalline iron oxide nanoparticle (MION), or the endogenous vascular-space-occupancy (VASO) mapping scheme to differentiate blood from brain tissue [20–23].

In contrast to the ASL-based orientation-specific flow measurement [24–27], phase contrast (PC) MRI relies on gradient-oriented dephasing of magnetized protons to map the velocity, i.e., direction and speed, of blood flow [28,29]. Past work with 7 T MRI showed that PC-MRI can be used to measure flow in the perforating arteries through the white matter and the lenticulostriate arteries in the basal ganglia of human brains [30–33]. However, the SNR was insufficient in these prior studies to map changes in flow, and thus changes in CBF, concurrent with changes in neuronal activation.

Here, we report on a PC-MRI method to detect vessel-specific changes in blood velocity on the timescale of single trials. We denote this method cerebral blood flow–related velocity, abbreviated CBFv. We build on past implementations of PC-MRI [30,34–37]. Here, we utilize a small surface RF coil with high-field MRI, i.e., 14.1 T, for improved SNR. This advance allows us to map the BOLD, CBV, and CBFv-fMRI signals from individual penetrating venules and arterioles, which span 20 to 70 μm diameter, with high spatial resolution [22,38,39].

## Results

### Phantom validation of high-resolution PC-based flow velocity measurement

For calibration, we constructed an in vitro capillary tubing circulatory system to mimic penetrating vessels, with flow rates from 1 to 10 mm/s (**Fig 1A**). A 2D PC-MRI slice is aligned perpendicular to the capillary tubing (**Fig 1A** and **1B**) and provides a voxel-specific measurement of the flow velocity through 2 tubes with the upward flow (positive sign, bright dots in **Fig 1B**) and 2 tubes with the downward flow (negative sign, dark dots in **Fig 1B**), as well as a control tube. We observe a monotonic and near linear relation between the velocity measured by PC-MRI and the true velocity: $V_{meas} = (0.67 \pm 0.01)\ V_{pump} + (0.02 \pm 0.11)$ mm/s at echo time (TE) = 5.0 ms (**Fig 1C**). The small offset could be caused by eddy current effects and other gradient-related scaling errors of the PC-MRI sequence [40–42]. We further observe that the measured velocities are relatively insensitive to the value of TE (**Fig 1C**).

We implemented the high-resolution PC-MRI with 14.1 T MR scanning for in vivo measurement of blood flow from individual penetrating arterioles and venules. We collected data

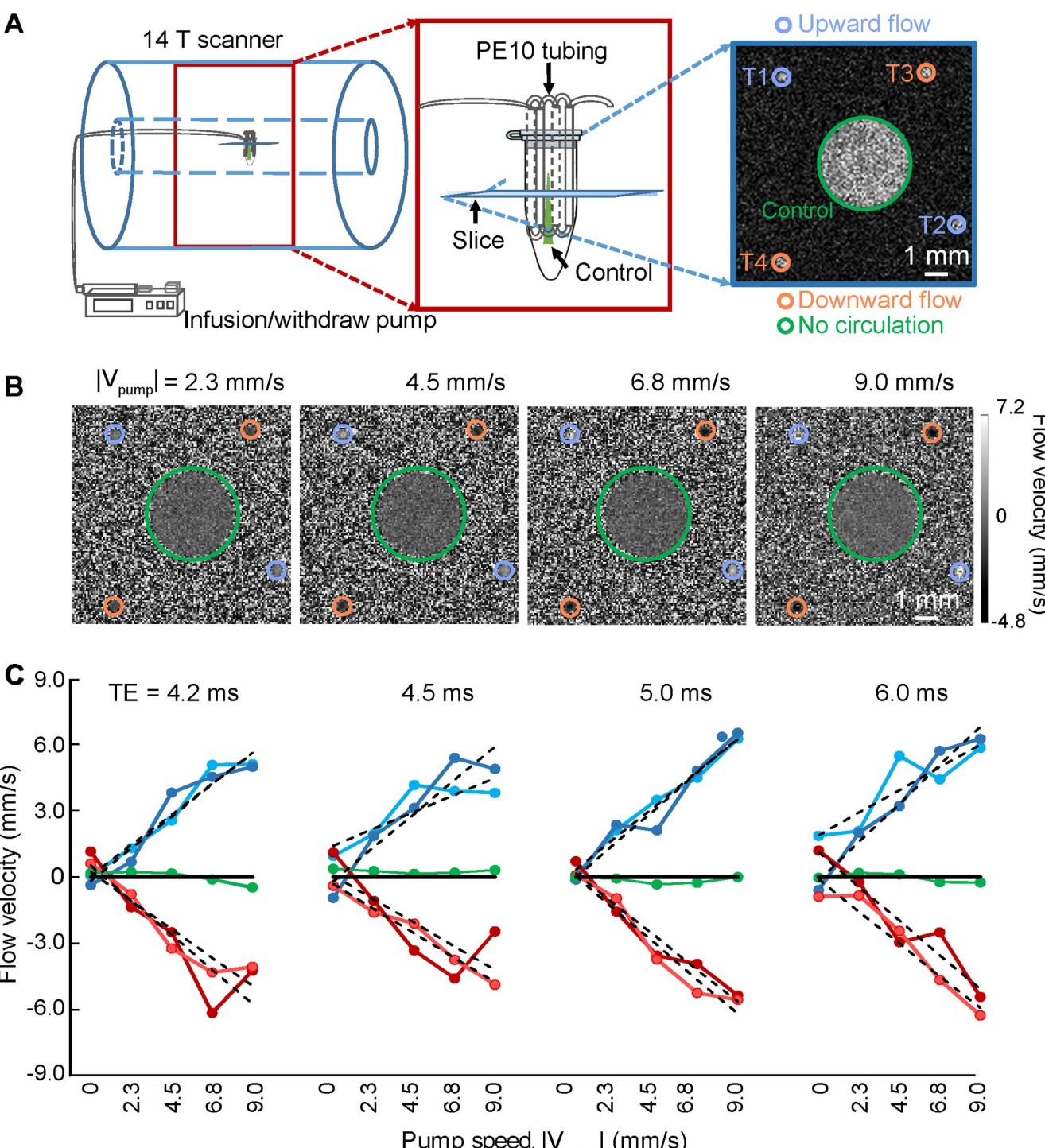

**Fig 1. In vitro flow velocity measurements with PC-MRI. (A)** Schematic drawing of the phantom experimental flow chamber in the 14.1 T scanner. An expanded image (red box) shows the circulatory system constructed of capillary tubes. A representative FLASH MRI image (blue box), 500 μm in thickness, shows the capillary positions. ROIs T1 and T2, contoured in purple, indicate the upward flow. ROIs T3 and T4, in orange contour, indicate the downward flow. The green contour indicates the stagnant fluid. **(B)** Representative images with different flow velocity in the capillaries T1 to T4 in panel A. TE = 5.0 ms for all panels. **(C)** The plot of flow velocity estimates from the 5 ROIs with different TEs, as marked, and different pump rates, as indicated and marked in panel B. The dotted lines correspond to a linear fitting for velocity measurements of different ROIs. The data underlying this figure can be found in S1 Data. FLASH, fast low angle shot; PC, phase contrast; ROI, region of interest; TE, echo time.

throughout the infragranular cortex, i.e., layer V, of anesthetized rats. A surface RF transceiver coil with 6-mm diameter was constructed and attached to the rat skull (**S1 Fig**) to improve the SNR of PC-MRI images as well as multi-gradient echo (MGE) images used for arteriole–venule (A–V) mapping [22,38]. This coil was essential for the high-resolution mapping with a fast sampling rate of the single-vessel flow velocity over a complete hemisphere of the rat brain (**Figs 2A and S1E**).

### In vivo PC-based flow velocity mapping of penetrating microvessels

We first acquired the single-vessel A–V map by aligning a 500-µm thick 2D MRI slice perpendicular to penetrating vessels through layer V of one hemisphere (**Fig 2A** and **2B**). We designed the pulse sequence for PC-MRI to achieve the same slice geometry of the A–V map

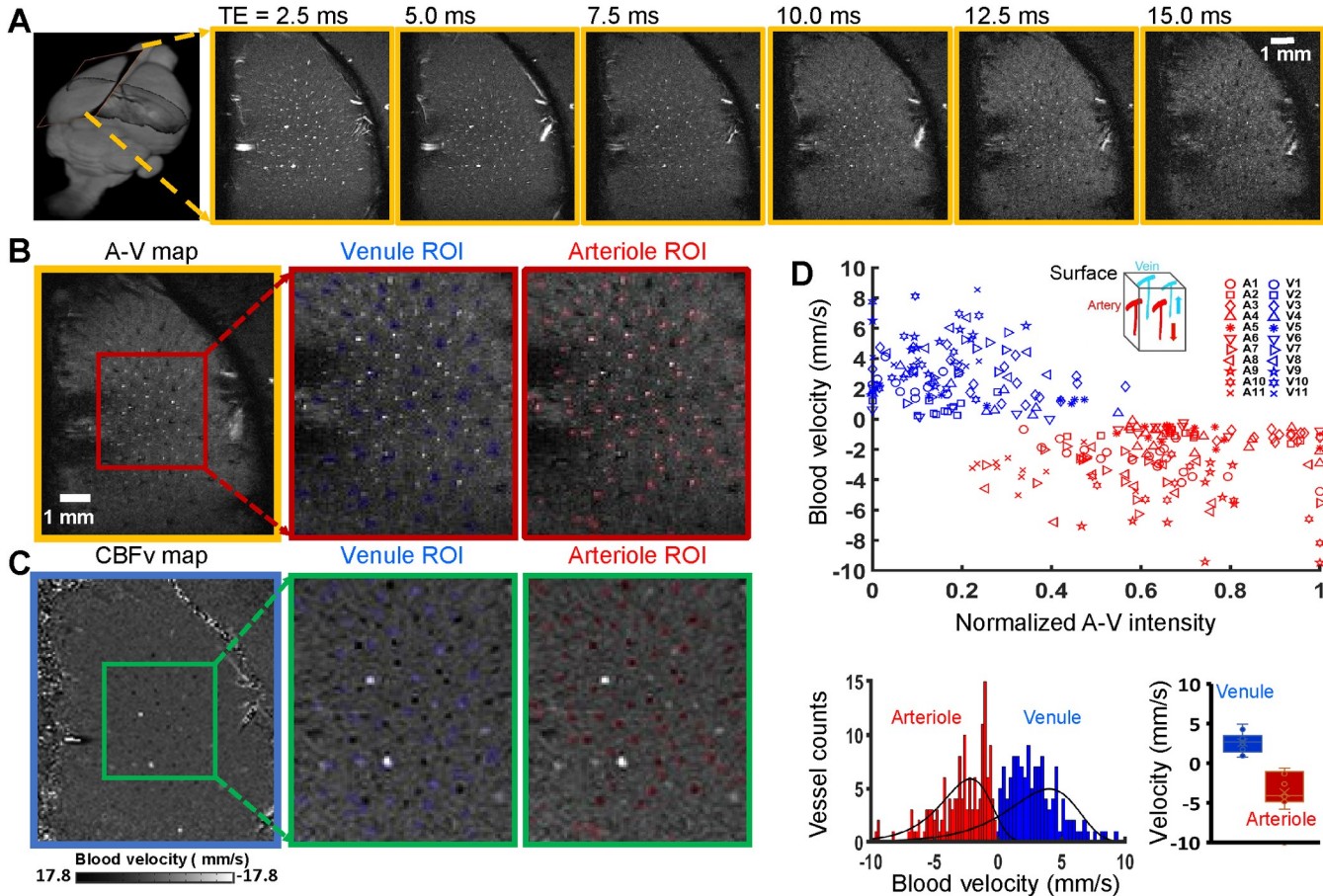

**Fig 2. PC-based single-vessel flow velocity (CBFv) mapping. (A)** Representative 2D MGE slices (yellow boxes) from a deep layer of the primary forepaw somatosensory cortex (first frame) at different TEs, as indicated. **(B)** The 2D A–V map (yellow box) derived from the images with different TEs in panel A, arterioles, and venules appear as bright and dark voxels, respectively. The expanded views (red boxes) show individual venules, i.e., black voxels marked in blue, and arterioles, i.e., white voxels marked in red. **(C)** The vectorized flow velocity map (blue box) from the same 2D MGE slice in panel B. The expanded views (green boxes) show the individual venules, i.e., white dots with positive velocity, and arterioles, i.e., black dots with negative velocity. Note that 2 bright dots are caused by the "overflowed" velocity beyond the maximal velocity, i.e., the Venc parameter, defined in the PC-MRI sequence, which could be not correctly estimated. **(D)** Scatter plot of the flow velocities from individual arterioles and venules as the function of the normalized signal intensities of each vessel in the A–V map of panel B, data from 11 rats as indicated. Insert shows the blood flow direction of arterioles and venules in the forepaw somatosensory cortical region. The lower panel shows the histogram of the blood velocity distribution across arterioles and venules, as well as the bar graph to show the mean velocity. The data underlying this figure can be found in S2 Data. A–V, arteriole–venule; CBFv, cerebral blood flow–related velocity; MGE, multi-gradient echo; PC, phase contrast; ROI, region of interest; Venc, velocity encoding; TE, echo time.

so that the CBF deduced from PC-MRI signals could be overlaid with individual penetrating arterioles and venules in the single-vessel flow velocity map (**Fig 2A** and **2C**). The arteriole blood flows into the cortex, while the venule blood flows outward, which determines the sign of the flow velocity. Vessel-specific velocities were plotted as a function of the normalized signal intensity in the A–V map and corroborated our ability to determine flow velocity specific to arterioles and venules (**Fig 2C** and **2D**). The measured flow velocities range from 1 to 10 mm/s, as previously measured with optical methods [43,44]. To probe the reliability of the single-vessel MR-based flow velocity method, we compared the velocities detected by PC-MRI methods with different TEs and flip angles and observed comparable results across a range of parameters (**S2 Fig**). It should be noted that altered vessel velocities detected from the same animal depend on the vessel sizes and orientation angles relative to the 2D slice. Lastly, the large variability of blood velocity across different animals could be caused by the varied physiological states of animals under anesthesia, as well as by degraded gradient performance during high duty cycle PC-MRI; see optimization step in Methods. All told, these data demonstrate the feasibility of in vivo PC-based blood velocity mapping from individual penetrating arteriole and venules.

## PC-based CBFv-fMRI from individual arterioles and venules

We contrasted the complementary capabilities of PC-based CBFv-fMRI against the signals observed with the balanced steady-state free precession (bSSFP)-based single-vessel BOLD- and CBV-fMRI mapping method [38] (**Fig 3**). We first created an A–V map through the infragranular layers of the forepaw region of the primary somatosensory cortex (**Fig 3Ai**), followed by 2D-bSSFP to detect stimulus-induced changes in the single-vessel BOLD fMRI signal (**Fig 3Aii**). We next performed single-vessel PC-MRI flow velocity measurements to measure baseline flow in penetrating arterioles and venules, using $100 \times 100 \ \mu m^2$ in-plane resolution, a sampling rate of 4-second repetition time (TR) per image and the same geometry as the 2D-bSSFP method (**Fig 3Aiii**). Changes in CBFv upon stimulation overlapped with individual penetrating vessels in the A–V map (**Fig 3Aiv**). Lastly, we performed 2D-bSSFP for single-vessel CBV-fMRI mapping by intravenous injection of iron particles into the blood in the same animals (**Fig 3Av**). The BOLD fMRI signal is primarily detected from individual penetrating venules, while the CBV-weighted signal is mainly located at the individual penetrating arterioles (dark dots in **Fig 3Aii** with bright dots in **Fig 3Av**). In contrast, the CBFv-fMRI signal is observed in both the penetrating arterioles and venules (**Fig 3Aiv**).

The stimulus-evoked responses of all CBV, BOLD, and CBFv-fMRI signals were studied with an on/off block design (**Fig 3B–3E**). Group analysis shows that the positive BOLD signal from venule voxels is significantly higher than the arteriole-specific BOLD signal (**Fig 3C**). In contrast, the arteriole dilation leads to an earlier CBV-weighted negative fMRI signal, which is much stronger and faster than the signal from passive venule dilation (**Fig 3C**), as expected [45–47]. Group analysis shows the similar temporal dynamics of CBFv changes, but with different signs in arterioles and venules (**Fig 3D**). In contrast to the venule-dominated BOLD and arteriole-dominated CBV responses (**Fig 3C** and **3F**), the percentage changes in CBFv-fMRI is statistically indistinguishable between arterioles ($3.12 \pm 0.87\%$) and venules ($3.60 \pm 0.80\%$) (**Fig 3E** and **3F**). The voxel-wise hemodynamic changes of BOLD, CBV, and CBFv are illustrated in **S3 Fig** and **S1 Movie**.

## Discussion

Despite the plethora of existing tools developed for CBF measurements in both animal and human brains, it remains challenging to noninvasively detect the flow dynamics of

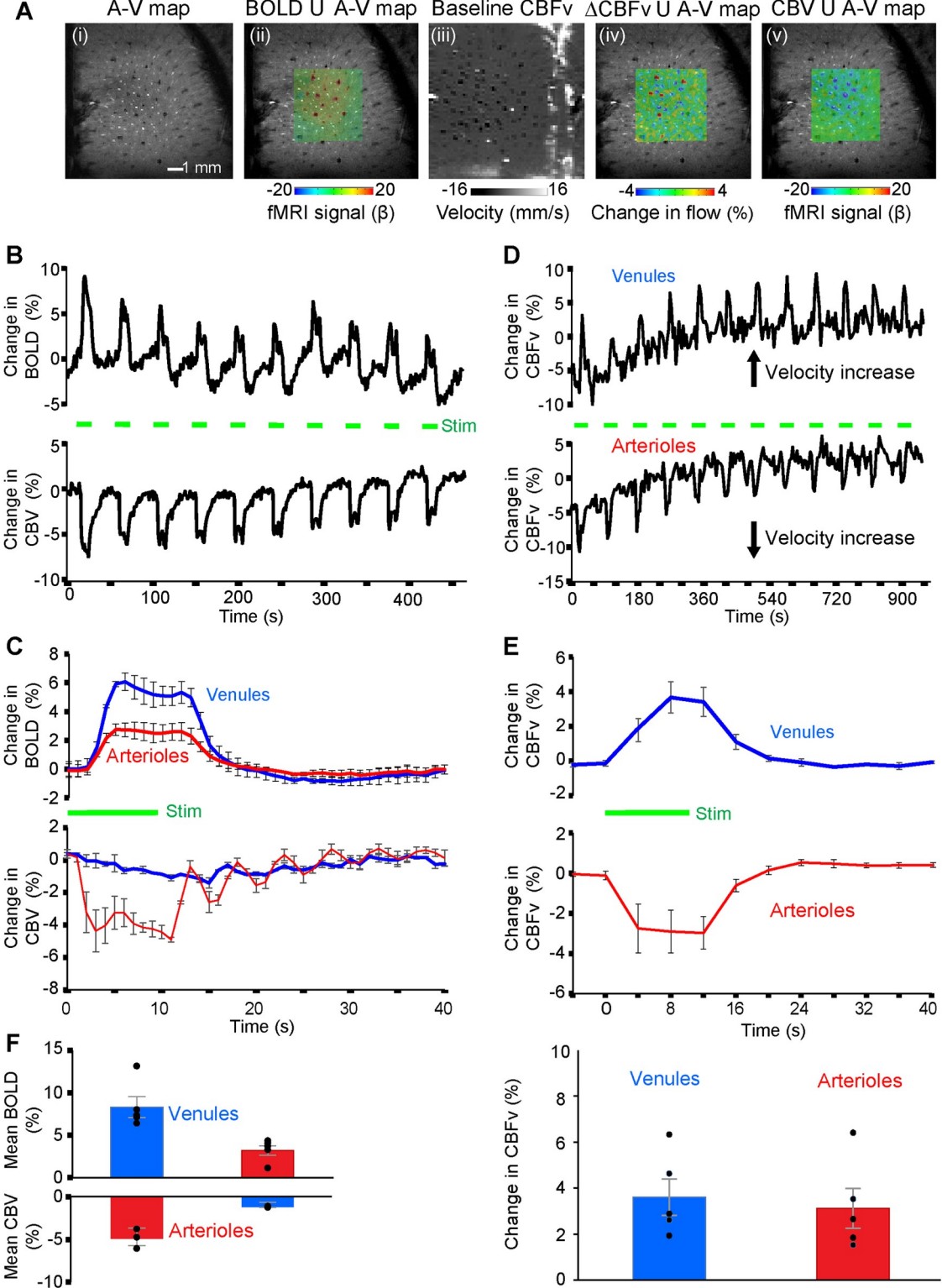

**Fig 3. Maps of task-related hemodynamic signals with single-vessel BOLD, CBV, and CBFv-fMRI. (A)** Different MRI measurement strategies on the same 2D slice. From left to right: (i) the A–V map defines arterioles as bright dots and venules as dark dots; (ii) the evoked bSSFP-based BOLD fMRI map, within a green subregion, on top of the A–V map; (iii) the PC-MRI map of baseline CBFv; (iv) the CBFv-fMRI map on top of the A–V map with an increased flow velocity corresponding to brighter voxels for venules and darker voxels for arterioles; and (v) evoked bSSFP-based CBV-fMRI map on top of the A–V map. **(B)** The time courses

of evoked bSSFP-BOLD and CBV-fMRI with the block design paradigm from venules and arterioles shown in panel A. Forepaw stimulation pulse of 330 μs in width and 1 mA in amplitude delivered at 3 Hz for 10 seconds. **(C)** Averaged time courses of the fractional change for evoked BOLD and CBV signals from venule and arteriole ROIs of different rats (mean ± SEM, the green bar shows stimulation duration). **(D)** The time courses of the evoked CBFv changes from the arteriole and venule ROIs show increased velocity from both arterioles and venules with the block design, 10-second duration stimulation paradigm. **(E)** The averaged time courses of the evoked CBFv changes show the velocity increase from both arteriole and venule ROIs with the block design stimulation paradigm from 5 rats (mean ± SEM, the green bar shows stimulation duration). **(F)** Bar graph shows that peak BOLD changes of venule are significantly higher than those of arteriole (5 rats, $p = 0.009$), while the peak CBV changes of arteriole are significantly higher than those of venule (3 rats, $p = 0.028$). In contrast, the peak CBFv changes are only slightly, but not significantly, higher in venules than arterioles (5 rats, $p = 0.063$). The data underlying this figure can be found in S3 Data. A–V, arteriole–venule; BOLD, blood oxygenation level–dependent; bSSFP, balanced steady-state free precession; CBFv, cerebral blood flow–related velocity; CBV, cerebral blood volume; ROI, region of interest.

intracortical microvessels. Here, we optimized PC-MRI to map the vectorized single-vessel flow velocity of penetrating arterioles and venules and further developed the single-vessel CBFv-fMRI technique to directly measure flow velocity in rat brains. In combination with previously established single-vessel BOLD- and CBV-fMRI methods, the PC-based single-vessel CBFv-fMRI method provides complementary information to map vessel-specific hemodynamic responses with high-resolution fMRI.

In contrast to the conventional ASL methods, PC-based MRI mapping allows arterioles and venules to be distinguished for simultaneous velocity measurements through a 2D plane. Also, ASL has less vascular specificity because water exchange through the blood–brain barrier of capillary beds increases the weighting of the ASL-based flow signal for parenchyma voxels [48,49]. Furthermore, there is significant variability in the transit time to flow from arterioles to venules through the capillary bed [50], which complicates the distinction of arterioles and venules by simple ASL-based CBF mapping. We detected the velocity from penetrating microvessels in the deep cortical layers with PC-MRI, with velocity values from 1 to 10 mm/s (**Fig 2**). It is noteworthy that the PC-based vessel velocity measurement is based on measuring water protons in blood but not limited to the flow of red blood cells. Still, the PC-based velocity from microvessels matches well with the previous optical measurement [2,43,44]. We conclude that high-resolution PC-MRI is ideal for noninvasive single-vessel CBFv-fMRI mapping.

The noninvasive measurement of microvessel blood velocity changes coupled to neuronal activation is a critical step to elucidate the neurovascular coupling mechanism underlying a variety of brain disorders in animal models. Existing methods, e.g., multiphoton microscopy [3,51,52] or doppler based ultrasound imaging [8–10], enable the detection of red blood cell velocity or CBF/CBV from microvessels with ultra-high resolution. Yet ultrasound requires a craniotomy or thinned skull [53] for effective spectrum-specific signal transmission through deep cortical layers or subcortical regions in rats and other animals. Such a surgical procedure has been well documented to induce perturbation of the physiological microenvironment in the brain [54–56], which potentially confounds the disease-related functional CBF measurements in animal brains. In contrast, conventional MRI methods, including PC-MRI, mainly detect blood flow or velocity in large vessels and seldom identify the functional CBFv of penetrating microvessels in the brain. By implementing a local RF coil with the high-field MR scanner, our work showed the feasibility to map the velocity changes from individual penetrating arterioles and venules with diameter 20 to 70 μm [22,38].

The novel applications of PC-based CBFv-fMRI can be divided into 2 different directions. First, the PC-based CBFv-fMRI provides a possible scheme to be combined with single-vessel BOLD and CBV fMRI signals for quantitative assessment of the cerebral metabolic rate of oxygen utilization in relation to flow [57]. These vessel-specific hemodynamic measurements can then be used to potentially validate the calibrated BOLD model [58] in future studies. Secondly, we now have a means to test the resting-state single-vessel correlation patterns of CBFv

signals from penetrating arterioles and venules (**S4 Fig**). In contrast to the $T2^*$-weighted resting-state readout of BOLD or CBV signal fluctuation [38], which amplifies the contrast-to-noise ratio (CNR) from the vessel-specific oxygenation or vessel diameter changes, CBFv signal fluctuation from penetrating microvessels provides a direct measurement of blood velocity changes across a large field of view. The measurement of PC-based CBFv signal fluctuation from microvessels can be further combined with single-vessel CBV-fMRI to study the vascular dynamic mechanism that underlies vasomotion-mediated perivascular clearance in animal models with Alzheimer disease (AD) [59,60]. In particular, both vasomotion-based ateriole diameter changes, i.e., the CBV signal fluctuation, and corresponding blood velocity changes, i.e., CBFv signal fluctuation, can be mapped across hippocampal penetrating vessels. This may be further combined with simultaneous fiber photometry based fluorescent recordings from genetically encoded biosensors [39] to identify the pathological vasodynamics in transgenic AD animals.

The accurate measurement of PC-based CBFv from microvessels relies on multiple factors. First, a remaining complication with PC-MRI mapping is the presence of small offsets in velocity as shown in our phantom capillary tubing studies with circulating flow under different conditions. The phase-dependent velocity encoding depends on the quality of the magnetic field gradients, and mismatched eddy currents of multiple gradients with opposite polarities, as well as the nonlinear and distorted gradient fields, could contribute to distortions in gradients [40–42]. We have described a list of parameters that should be optimized when implementing high-resolution PC-based CBFv measurements (Methods). Also, the high-resolution PC-MRI method is a high duty cycle sequence and slight heating of the gradient coil during scanning may alter the gradient performance, consistent with the baseline drift of the CBFv-fMRI signal in the first 5 minutes of scanning (**Fig 3D**). Nevertheless, it should be noted that the percentage velocity changes from individual arterioles and venules can be readily detected with the PC-based CBFv-fMRI measurement regardless of the gradient heating–related baseline drift. Another factor that contributes to the phase-dependent velocity error originates from the limited spatial resolution of PC-MRI images in comparison to the diameters of small vessels, i.e., the partial volume effect [36], although corrections are possible [30,37].

The present work shows the feasibility of PC-based CBFv-fMRI in rat brains using a ultra–high-field strength MR scanner (14 T). It should be noted that the high-resolution PC-based CBFv measurement can also be applied with a 9.4 T scanner to detect the microvessel blood velocity at a similar scale to 14 T measurement (**S5 Fig**). This supports a broader usage of non-invasive animal CBFv mapping. The translational potential of the PC-based CBFv-fMRI remains to be investigated. Previously, we detected low-frequency fluctuation of single-vessel resting state BOLD fMRI signals (TR = 1 second) from individual sulcus veins and arteries in the occipital lobe of the human brain [38]. The $T2^*$-weighted BOLD resting state fMRI signal fluctuation from individual vessels can be separated from noise artifacts due to pulsation or other motion effects. In contrast, pulsation could be a significant confounding issue for PC-based blood velocity mapping from individual vessels [61,62]. To better differentiate the pulsational contribution to CBFv signal fluctuation given different frequency ranges, we will need to increase the sampling rate by implementing phased-array surface coils for focal field of view measurement and an accelerated PC-MRI sequence.

## Methods

### Design of a phantom capillary tubing flow system

To validate the PC-MRI sequence, a plastic circulatory flow phantom composed of the capillary tubing (PE-10, Instech Laboratories, (Plymouth Meeting, PA, USA) inner diameter

210 μm) was constructed to mimic the geometry of cortical blood vessels (Fig 1A). The capillary tube was connected to a programmable syringe infusion/withdraw pump (Pump Elite 11, Harvard Apparatus, Holliston, MA, USA) with an infusion rate of 0.25, 0.5, 1.0, 1.5, and 2.0 mL/h, which were transferred to the flow velocity of the capillary tubing as shown in Fig 1. We infused manganese solution (50 mM $MnCl_2$, Sigma-Aldrich, Germany) through the tubing. The phantom capillary tubing was immersed in Fomblin (Sigma-Aldrich, Germany) to avoid air interface artifacts.

## Animal preparation for fMRI

All surgical and experimental procedures were approved by local authorities (Regierungspraesidium, Tübingen Referat 35, Veterinärwesen, Leiter Dr. Maas, protocol #KY 2/14 2/17) and were in full compliance with the guidelines of the European Community (EUVD 86/609/EEC) and guidelines of the Animal Care and Use Committee and the Animal Health and Care Section of Massachusetts General Hospital. The experimental animals were Sprague-Dawley male rats, approximately 250 g, provided by the Charles River Laboratories. A total of 25 rats were used in all experiments (the evoked bSSFP-BOLD/CBV and PC-MRI signals were acquired from 5 of 25 rats).

Detailed surgical procedures have been described previously [38,39]. Briefly, rats were first anesthetized with isoflurane, 5% (v/v) induction, and 1% to 2% (v/v) maintenance, and each rat was orally intubated with a mechanical ventilator (SAR-830, CWE, PA, USA). The femoral artery and vein were catheterized with plastic catheters (PE-50, Instech Laboratories) to monitor the arterial blood gas, administrate drugs, and constantly measure the blood pressure. After catheterization, rats were secured in a stereotaxic apparatus, and a custom-made RF coil was fixed above the skull with cyanoacrylate glue (454, Loctite, Henkel, Dusseldorf, Germany). After surgery, isoflurane was switched off and a bolus of α-chloralose (80 mg/kg, Sigma-Aldrich, Germany) was intravenously injected. A mixture of α-chloralose (26.5 mg/kg/h) and the muscle relaxant (pancuronium bromide, 2 mg/kg/h) was continuously infused to maintain the anesthesia and minimize motion artifacts. Throughout the whole experiment, the rectal temperature of rats was maintained at 37˚C by using a feedback heating system. All relevant physiological parameters were constantly monitored during imaging, including heart rate, rectal temperature, arterial blood pressure, the pressure of the tidal ventilation, and end-tidal $CO_2$. Arterial blood gases were checked to guide the physiological status adjustments by changing the respiratory volume or administering sodium bicarbonate (NaBic 8.4%, Braun, Melsungen, Germany) to maintain normal pH levels. Dextran-coated iron oxide particles (15 to 20 mg of Fe/kg, BioPAL, Massachusetts, USA) were intravenously injected for CBV-weighted signal acquisition.

## fMRI setup

All images were acquired with a 14.1 T, 26 cm horizontal bore magnet (Magnex Scientific, Oxford, UK) interfaced through the Bruker Advance III console (Bruker, Billerica, Massachusetts, USA). The scanner is equipped with a 12-cm magnet gradient set capable of providing a strength of 100 G/cm and a 150 μs rise time (Resonance Research, Billerica, MA, USA). We also applied the 9.4 T scanner (Bruker) with a 21-cm bore size (Magnex Scientific) to test the PC-MRI sequence. A custom-made transceiver coil with an internal diameter of 6 mm was used for fMRI image acquisition. For the electrical stimulation, 2 custom-made needle electrodes were placed on the forepaw area of the rats to deliver the electrical pulse sequences; 330 μs duration at 1.0 ~ 2.0 mA repeated at 3 Hz for 10 seconds using a stimulus isolator (A365, WPI, Sarasota, FL, USA). The stimulation duration and frequency were triggered

directly through the MRI scanner which was controlled by Master-9 A.M.P.I. system (Jerusalem, Israel). The triggering pulses from the MRI scanner were also recorded by the Biopac system (MP150, Biopac Systems, USA).

## Single-vessel MGE imaging

To anatomically map individual arterioles and venules penetrating deep cortical layers of the somatosensory cortex, a 2D MGE sequence was applied with the following parameters: TR = 50 ms; TE = 2.5, 5.0, 7.5, 10.0, 12.5, and 15.0 ms; flip angle = 55˚; matrix = 192 × 192; in-plane resolution = 50 × 50 μm$^2$; slice thickness = 500 μm. The A–V map was made by averaging MGE images from the second echo to the fifth echo. In the A–V map, the arteriole voxels show bright (red marks) due to the in-flow effect and venule voxels show as dark dots (blue marks) because of the fast $T_2^*$ decay (**Fig 2B**).

## bSSFP BOLD- and CBV-fMRI

The bSSFP sequence was applied to acquire the evoked BOLD signals by using the following parameters: TR = 15.6 ms; TE = 7.8 ms; flip angle = 15˚; matrix = 96 × 96; FOV = 9.6 × 9.6 mm$^2$; in-plane resolution = 100 × 100 μm$^2$; slice thickness = 500 μm. For the bSSFP CBV-fMRI, the parameters were adjusted with TR = 10.4 ms and TE = 5.2 ms. The total TR to acquire each image is 1 second. To reach the steady state, 300 dummy scans were used, followed by 25 pre-stimulation scans, 1 scan during stimulation, and 44 post-stimulation scans with 10 epochs for each trial. The fMRI stimuli block design of each trial consisted of 10-second stimulation and 35-second interstimulus interval. The total acquisition duration of each trial was 7 minutes 55 seconds. CBV-weighted fMRI signals were acquired after intravenous injection of dextran-coated iron oxide particles (15 ~ 20 mg of Fe/kg, BioPAL).

## PC-MRI

To measure the flow velocity of individual arterioles and venules, the PC-MRI sequence was applied with the following parameters. For the in vitro phantom and in vivo single-vessel CBFv measurement: TR = 15.6 ms; TE = 4.2, 4.5, 5.0, 6.0 ms; flip angle = 25˚; FOV = 6.4 × 6.4 mm$^2$; matrix = 128 × 128; in-plane resolution = 50 × 50 μm$^2$; slice thickness = 500 μm; maximum velocity encoding (Venc) = 0.66 to 1.56 cm/s (based on the flow values). The total acquisition time was 11 minutes 28 seconds. For the in vivo CBFv-fMRI measurements: TR = 15.6 ms; TE = 5 ms; flip angle = 30˚; FOV = 6.4 × 6.4 mm$^2$; matrix = 64 × 64; in-plane resolution = 100 × 100 μm$^2$; slice thickness = 500 μm. A total TR for each image is 4 seconds. The total acquisition duration of each trial was 16 minutes. The total acquisition duration of each trial was 16 minutes. For in vivo single-vessel CBFv measurement with 9.4T scanner: TR = 15.6 ms; TE = 5.0 ms; flip angle = 15˚; FOV = 6.4 × 6.4 mm$^2$; matrix = 96 × 96; in-plane resolution = 67 × 67 μm$^2$; slice thickness = 500 μm; maximum Venc = 0.91 to 1.58 cm/s (based on velocity values). To measure the blood flow velocity, bipolar flow encoding gradients were applied along the slice encoding direction. The slice position was anatomically identical with the slice position of the MGE imaging.

## Parameter optimization for the single-vessel CBFv mapping

(i) The orientation of the 2D slice should be parallel to the x-z plane of the magnet when performing slice-oriented velocity measurement, which would reduce cross-axis gradient-dependent error. (ii) The Venc value needed to be matched with the maximal velocities likely to be encountered within the vessel of interest. Since the Venc value is inversely proportional to the

bipolar gradient, the minimal Venc value depends on the gradient strength. It should be noted that too low Venc values could also lead to gradient heating artifacts due to the high duty cycle. (iii) We usually applied high-order MapShim to improve the field homogeneity before the high-resolution PC-MRI. Meanwhile, to remove field homogeneity-related phase offset, a pair of toggled bipolar gradients was typically applied. It is important to note that the mismatched amplitudes of toggled bipolar gradient amplitudes or related eddy currents could lead to background velocity error. We could manually calibrate the appropriate gradient amplitude using the build-in parameters of the Bruker system, i.e., PVM_PPGradAmpArray.

### Data analysis and statistics

All data preprocessing and analysis were performed by using the software package, Analysis of Functional NeuroImages (AFNI) (NIH, Bethesda, USA). All relevant fMRI analysis source codes can be downloaded from (https://afni.nimh.nih.gov/pub/dist/doc/htmldoc/ background_install/main_toc.html).

### Definition of the individual vessels

The individual arteriole/venule voxels were defined by the signal intensity of the A–V map [22]. The arterioles are determined if the voxel intensities are higher than the mean signal intensities plus 2 times the standard deviation of the local area in a $5 \times 5$ kernel. The venules are determined if the voxel intensities are lower than the mean signal intensities minus 2 times the standard deviation of the local area [22,38,39]. The locations of individual arteriole/venule voxels defined in the A–V map were used to extract the time courses of BOLD/CBV-fMRI for individual vessels.

### BOLD/CBV-fMRI and PC-MRI data analysis

To register the evoked bSSFP-fMRI images and evoked PC-MRI images with the 2D anatomical A–V map, the tag-based registration method was applied. A total of 12 to 15 tags were selected from the averaged bSSFP-fMRI images or the averaged PC-fMRI images to register those selected from the A–V map. We used a 3dLocalstat AFNI function to normalize the signal intensity of the single-vessel maps. This process allowed us to plot the PC-based velocity values of individual vessel voxels to the normalized signal intensity of A–V maps. For the evoked signals, the bSSFP-fMRI images and PC-MRI images were normalized by scaling the baseline to 100. Multiple trials of block design time courses were averaged for each animal. No additional smoothing step was applied. The β-value was calculated to measure the amplitude of the fMRI responses at each TR. The voxel-wise β map was illustrated with the spatial pattern of the fMRI responses corresponding to the different time points after the stimulus onset. After registration (tag-based registration) and region of interest extraction (3dLocalstat function, mask shown in **Fig 2B**), we extracted the PC-based flow velocity values from individual vessel voxels, which were identified based on the algorithm as described in the previous section.

The hemodynamic response function is based on the "block function" of 3dDeconvolve module developed in AFNI. The HRF model is defined as follows:

$$h(t) = \int_0^{\min(t,L)} s^4\, e^{-s}/[4^4 e^{-4}]\, ds$$

Gamma variate function = $s^4 e^{-s} / 4^4 e^{-4}$. L was the duration of the response. BLOCK (L, 1) is a convolution of a square wave of duration L that makes a peak amplitude of block response = 1.

For the resting-state single-vessel CBFv-fMRI correlation analysis, the seed-based voxel-wise correlation map was created using AFNI. The Welch power spectrum density plot was calculated from arteriole and venule voxels with averaged periodogram method (a 240-second timing window into overlapping sections with 60-point and 25% overlap). The sampling rate of rs-CBFv-fMRI signal is 4 seconds, allowing the analysis of low-frequency signal oscillation less than 0.125 Hz.

For the group analysis, Student $t$ test was performed, and error bars are displayed as the means ± SEM. The $p$-values $< 0.05$ were considered statistically significant. The sample size of animal experiments is not previously estimated. No blinding and randomization design was needed in this work.

## Supporting information

**S1 Fig. The preparation of in vivo experiment for the PC-MRI in 14.1 T. (A)** The flowchart of the in vivo experiment in the 14.1 T scanner. **(B)** Photograph of the custom-made transceiver surface RF coil. **(C)** Photograph of the coil position: The coil is glued to the rat skull. **(D)** The schematic drawing of the rat position inside the MRI holder. **(E)** Representative images from different views of the FLASH MRI show the ideal coil position. FLASH, fast low angle shot; PC, phase contrast; RF, radio frequency.
(TIFF)

**S2 Fig. Phase images from the different representative rats with different TEs and flip angles. (A)** Phase images from a representative rat with different TEs, i.e., 2.75, 3.0, and 3.2 ms. The right panel shows the mean blood flow velocity (mean ± SEM) from left images with $N_{Arteriole} = 48$ and $N_{Venule} = 22$. **(B)** Phase images from a representative rat with different flip angles, i.e., 25˚, 30˚, and 35˚. The right panel shows the mean CBFv from left images with $N_{Arteriole} = 38$ and $N_{Venule} = 14$. The data underlying this figure can be found in S4 Data. CBFv, cerebral blood flow–related velocity; TE, echo time.
(TIFF)

**S3 Fig. The bSSFP-based single-vessel BOLD/CBV-fMRI and the PC-based single-vessel CBFv measurement from a representative rat. (A)** The evoked bSSFP-based BOLD- (left) and CBV- (right) fMRI maps overlaid on the A–V map of a representative rat, with the voxel-wise time courses from the ROIs of individual venule and arteriole voxels (10 seconds on and 35 seconds off for 10 epochs plotted in a $3 \times 3$ matrix). **(B)** The evoked CBFv functional maps overlaid on the A–V map of a representative rat. The voxel-wise time courses of CBFv changes from the same ROIs of individual venule and arteriole voxels (10 seconds on and 50 seconds off for 12 epochs plotted in a 3 x 3 matrix). A–V, arteriole–venule; BOLD, blood oxygenation level–dependent; bSSFP, balanced steady-state free precession; CBFv, cerebral blood flow–related velocity; CBV, cerebral blood volume; PC, phase contrast; ROI, region of interest.
(TIFF)

**S4 Fig. The PC-based single-vessel resting-state CBFv-fMRI mapping. (A)** The A–V map shows bright dot as arterioles and dark dots as venules from one representative rat. Seed-based correlation maps were overlapped on the A–V map, showing 2 venule seeds (V1 and V2) and 2 arteriole seeds (A1 and A2). The venule-based correlation map shows positive correlations to other venules, but negative correlations with arterioles. In contrast, the arteriole-based correlation map shows positive correlations to other arterioles, but negative correlations with venules.

**(B)** The enlarged CBFv map shows the venule with positive velocity (bright dots) and arterioles with negative velocity (dark dots). The venule seed-based CBFv correlation map was over-lapped on the CBFv map, showing the positive correlation on surrounding venule voxels (bright dots), and negative correlation on surrounding arteriole voxels (dark dots). **(C)** The normalized time course extracted from venule and arteriole voxels, showing correlated low-frequency signal fluctuation. **(D)** The PSD plot of resting state CBFv dynamics from arteriole and venule voxels shows the slightly higher power from 0.01 to 0.04Hz ($n = 5$). The data under-lying this figure can be found in S5 Data. A–V, arteriole–venule; CBFv, cerebral blood flow–related velocity; PC, phase contrast; PSD, power spectrum density.
(TIFF)

**S5 Fig. The PC-based CBFv mapping in rats with the 9.4 T scanner. (A)** The vectorized flow velocity (CBFv) map with $67 \times 67$ um$^2$ in-plane resolution shows venules (bright dots) with positive CBFv values and arterioles (dark dots) with negative CBFv values. **(B)** The bar graph shows the averaged velocity for arterioles ($-1.37 \pm 0.94$ mm/s) and venules ($1.02 \pm 0.32$ mm/s) from 4 rats. The data underlying this figure can be found in S6 Data. CBFv, cerebral blood flow–related velocity; PC, phase contrast.
(TIFF)

**S1 Movie. The bSSFP-based single-vessel BOLD/CBV-fMRI and the PC-MRI based single-vessel CBFv-fMRI in the rat cortex.** The left panel shows the evoked bSSFP-based BOLD- (upper) and CBV- (lower) fMRI responses with a representative trace from single venule and arteriole voxels (arrow) (10 seconds on and 35 seconds off for a total of 45-second time window with TR = 1 second). The right panel shows the evoked PC-based CBFv-fMRI responses with a representative trace from single venule (upper) and arteriole (lower) voxels (10 seconds on and 50 seconds off for a total of 60-second time window with TR = 4 seconds). Note that the underlay is the A–V map showing venules as dark voxels and arterioles as bright voxels. A–V, arteriole–venule; BOLD, blood oxygenation level–dependent; bSSFP, balanced steady-state free precession; CBFv, cerebral blood flow–related velocity; CBV, cerebral blood volume; PC, phase contrast.
(MP4)

**S1 Data. The flow velocity estimates from the 5 ROIs with different TEs and different pump rates.** The flow velocity estimates from the 5 ROIs with different TEs (4.2, 4.5, 5.0, and 6.0 ms) and different pump rates (0, 2.3, 4.5, 6.8, and 9 mm/s). The linear fitting is calculated from averaged velocity of venules and arterioles. ROI, region of interest; TE, echo time.
(XLSX)

**S2 Data. Scatter plot of the flow velocities from individual arterioles and venules as the function of the normalized signal intensities of each vessel in the A–V map from 11 rats. Fig 2D upper panel:** The flow velocities from individual arterioles and venules as the function of the normalized signal intensities of each vessel in the A–V map. The raw data was acquired from 11 animals. **Fig 2D lower panel:** Mean velocity of arterioles and venules for the bar graph from 11 animals. **MATLAB code and mat file:** The MATLAB code and raw data for the histogram of the blood velocity distribution across arterioles and venules from the animals. A–V, arteriole–venule.
(RAR)

**S3 Data. Hemodynamic signal data for single-vessel BOLD, CBV, and CBFv-fMRI measurement. Fig 3B:** The time courses of evoked bSSFP-BOLD and CBV-fMRI with the block design stimulation paradigm from venules and arterioles. **Fig 3C:** The averaged time courses

of the fractional change for evoked BOLD and CBV signals from venule and arteriole ROIs and the raw data from 3 animals. **Fig 3D:** The time courses of the evoked CBFv changes from the arteriole and venule ROIs from both arterioles and venules with the block design stimulation paradigm. **Fig 3E:** The averaged time courses of the evoked CBFv changes from arteriole and venule ROIs with the block design stimulation paradigm and the raw data from 5 animals. **Fig 3F:** The venules and arterioles ROIs peak signals for BOLD, CBV, and CBFv measurements from different animals. BOLD, blood oxygenation level–dependent; bSSFP, balanced steady-state free precession; CBFv, cerebral blood flow–related velocity; CBV, cerebral blood volume; ROI, region of interest.
(XLSX)

**S4 Data. Blood flow velocity with different TEs and flip angles. S2A Fig:** The blood flow velocity of 48 arterioles and 22 venules with different TEs (2.75, 3.0, and 3.2 ms). **S2B Fig:** The blood flow velocity of 38 arterioles and 14 venules with different flip angles (25˚, 30˚, 35˚). TE, echo time.
(XLSX)

**S5 Data. The PC-based single-vessel resting-state CBFv-fMRI PSD.** The PSD of resting state CBFv dynamics from arteriole and venule voxels from 5 animals. CBFv, cerebral blood flow–related velocity; PC, phase contrast; PSD, power spectrum density.
(XLSX)

**S6 Data. The PC-based CBFv mapping in rats with the 9.4 T scanner.** The averaged velocity for arterioles and venules from 4 animals. CBFv, cerebral blood flow–related velocity; PC, phase contrast.
(XLSX)

## Acknowledgments

We thank Dr. E. Weiler and Ms. S. Fischer for animal/lab maintenance and support, Dr. N. Avdievitch for technical support, and the Analysis of Functional NeuroImages (AFNI) team for their software support.

## Author Contributions

**Conceptualization:** David Kleinfeld, Xin Yu.

**Data curation:** Xuming Chen, Yuanyuan Jiang, Xin Yu.

**Formal analysis:** Xuming Chen, Yuanyuan Jiang.

**Funding acquisition:** Xin Yu.

**Methodology:** Xuming Chen, Yuanyuan Jiang, Rolf Pohmann, Klaus Scheffler, Xin Yu.

**Project administration:** Xin Yu.

**Resources:** Xin Yu.

**Supervision:** Xin Yu.

**Writing – original draft:** Xuming Chen, David Kleinfeld, Xin Yu.

**Writing – review & editing:** Yuanyuan Jiang, Sangcheon Choi, David Kleinfeld, Xin Yu.

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
