## [Editor Report · Decision Letter 0]

28 Aug 2020

Dear Dr Yu, 

Thank you for submitting your manuscript entitled "Single-vessel cerebral blood flow fMRI to map blood velocity by phase-contrast imaging" for consideration as a Methods and Resources by PLOS Biology.

Your manuscript has now been evaluated by the PLOS Biology editorial staff as well as by an academic editor with relevant expertise and I am writing to let you know that we would like to send your submission out for external peer review.

Please re-submit your manuscript within two working days, i.e. by Aug 30 2020 11:59PM.

Kind regards,

Lucas Smith, Ph.D.,

Associate Editor

PLOS Biology

---

## [Decision Letter · Decision Letter 1]

29 Oct 2020

Dear Dr Yu,

Thank you very much for submitting your manuscript "Single-vessel cerebral blood flow fMRI to map blood velocity by phase-contrast imaging" for consideration as a Methods and Resources at PLOS Biology. Your manuscript has been evaluated by the PLOS Biology editors, an Academic Editor with relevant expertise, and by several independent reviewers.

The reviews of your manuscript are appended below. As you will see from their comments, the reviewers have raised a number of specific points that need to be addressed to strengthen the study before we can consider it for publication at PLOS Biology. For example, reviewer 1 would like additional clarifications on the velocity measurements and reviewer 2 has noted potential limitations of the PC-MRI approach that are not addressed. Importantly, reviewer 2 considers that the manuscript does not currently demonstrate the advantage of the PC-MRI approach over existing methods for either basic or clinical researchers, or demonstrate the potential to experimentally address a previously inaccessible biological question. These are requirements for PLOS Biology Methods and Resources articles. A revised manuscript would need to demonstrate an advantage over existing methods and/or how it can be implemented to gain previously elusive biological insights to make a strong case for publication here.

In light of the reviews, we will not be able to accept the current version of the manuscript, but we would welcome re-submission of a much-revised version that takes into account the reviewers' comments. We cannot make any decision about publication until we have seen the revised manuscript and your response to the reviewers' comments. Your revised manuscript is also likely to be sent for further evaluation by the reviewers

We expect to receive your revised manuscript within 3 months. However, please let us know if you would like an extension, as it is important that the revised manuscript thoroughly addresses the reviewer concerns. 

**IMPORTANT - SUBMITTING YOUR REVISION**

*Re-submission Checklist*

*Published Peer Review*

*PLOS Data Policy*

*Blot and Gel Data Policy*

Sincerely,

Lucas Smith, Ph.D.,

Associate Editor,

lsmith@plos.org,

PLOS Biology

REVIEWS:

Reviewer's Responses to Questions

PLOS authors have the option to publish the peer review history of their article (what does this mean?). If published, this will include your full peer review and any attached files.

Reviewer #1: No

Reviewer #2: No

Reviewer #3: Yes: Cornelius Faber

Reviewer #1: This paper is a follow up from their previous work which elegantly demonstrated single vessel BOLD and CBV -fMRI. Here they extend this platform to include single vessel velocity measurements using high-resolution phase-contrast MRI. This is a powerful technique with potential applications towards high-resolution imaging in humans and understanding hemodynamic responses in animal models. The authors capture single vessel signals for BOLD, CBV and velocity across single trials. I am enthusiastic about the study, but I do have a couple concerns.

1. Flow and velocity are intermixed throughout the manuscript which is confusing. I would prefer the authors avoid the use of flow to describe velocity. CBF is the quantity/time (eg. ml/min, RBCs/second) whereas velocity is the distance/time. I also suggest changing the name from CBF-fMRI to something else.

2. Upon stimulation, it is surprising that the velocity seems to increase equally in both the arteriole and the veins. Due to the arteriole specific volume increase, I would expect a smaller velocity change in the arteriole than the veins. Previous work in olfactory bulb (PMID: 29937277), demonstrated such a phenomenon with two-photon measurements of vessel diameter and velocity. Do the authors ever see mismatches between volume and velocity with their technique? 

3. Can the authors be confident the technique accurately reports absolute velocity? The velocities they report seem very slow when compared to previous measurements of arteriole RBC velocity with two-photon scanning, (e.g. PMID: 19174826, Fig 1). From Fig 2D it appears single arteriole velocities of >3mm/s were only observed in 1 of 6 rats and no arterioles had velocities above 5mm/s. In Figure S2 Rat#2 has a mean arteriole velocity of <0.5mm/s (38 arterioles). Could there be contamination from tissue parenchyma, interstitial fluid, and surrounding capillary bed? The authors state possible partial volume effects in the last sentence of the discussion. But it would be important to clearly state how this may affect the quantitative measurements being made to avoid misinterpretation of the results. 

Minor: 

Fig 2C, could the authors add a velocity scale bar. 

L97 - typo, "while" matter

Reviewer #2: This manuscript describes the use of phase contrast MRI (PC-MRI) for detecting velocity and velocity changes in signal cortical arteries and veins in the rat brain. High resolution PC MRI was performed at 14.1 T in anesthetized rats and compared to BOLD fMRI and contrast-enhanced CBV based fMRI for detecting functional activation during forepaw stimulation. Whereas BOLD fMRI was more sensitive to venules than arterioles and CBV MRI was more sensitive to arterioles than venules, PC-MRI sensitivity was comparable between venules and arterioles. The authors conclude that the method will be useful for studying disease models and can be translated to human use for studying vascular dementia and other brain disorders. While the methods and resulting data are reasonable, the main novelty of the work is carrying out PC-MRI at higher resolution than has been used previously while the rationale and implications of the method are poorly developed and its limitations are inadequately discussed. The manuscript could be improved by consideration of the following issues, all of which are essential to improving the work:

1. The first sentence of the introduction describes CBF as a "key readout of neuronal processing." That is not strictly correct. Through neurovascular coupling, CBF is coupled to neural and not neuronal activity. It is less clear whether neuronal processing (spike patterns, etc.) are tightly linked to CBF changes.

2. CBF generally refers to a quantifiable metric of flow, either in ml/min or ml/g/min. In the current work and much of the first paragraph of the introduction, the authors are referring to flow or changes in flow and not quantified CBF. For example, NIRS certainly does not measure CBF. It measures changes in oxyhemoglobin/deoxyhemoglobin ratios that may reflect underlying changes in CBF.

3. The discussion of flow sensitive methods leaves out vascular space occupancy (VASO) contrast.

4. (line 95) The authors are incorrect that ASL cannot show orientation-specific effects. Because continuous arterial spin labeling methods leverage directional flow, they contain directional information (see e.g. https://pubmed.ncbi.nlm.nih.gov/26968145/)

5. (lines 141-1) All PC MRI is single vessel, so this sentence should be modified to contain information about the type of vessels.

6. (line 153) The manuscript describes results on "stimulation" without describing the nature of the stimulus. Although that is covered in the Methods, given the organization of the manuscript with the methods at the end, the Results should be more descriptive.

7. Figure 3D is never described in the text of the manuscript. Figure 3E uses a different timing on the x-axis from 3C, which makes comparisons more difficult than it needs to be. The authors might also consider flipping the CBV and arteriolar plots in Figures B-E along the y-axis so that the changes show the same polarity as the other plots.

8. The supplemental movie is also difficult to follow.

9. Although a number of rats were studied with multiple contrast mechanisms, there does not appear to be any statistical treatment of the data.

10. The authors state in the Discussion and elsewhere that they have "optimized" PC-MRI in this work, but no optimization is presented.

11. One of the main weaknesses of this manuscript is the failure to address limitations of the PC-MRI approach. Firstly, because velocity encoding is directional in PC-MRI, it is sensitive to the angle of the vessel with respect to the velocity encoding gradient. Accordingly, the measured velocity changes will vary with this angle. While there may be literature demonstrating that cortical microvessels are largely orthogonal to the cortical surface, which could minimize such errors, this issue should be addressed. Even if it is true that cortical microvessels are orthogonal to the surface, the cortical surface is curved and the imaging slab is not. A second issue is that while imaging resolution is very high (50 microns) in-plane, the imaging slab is an order of magnitude larger. So, there are also likely significant partial volume effects in this approach. Both of these limitations undermine accurate quantification of flow changes.

12. Perhaps the biggest weakness of the manuscript is the failure to address the significance of PC-MRI versus existing methods for studying cortical function. While the authors demonstrate that PC-MRI is roughly equally sensitive to both arteriolar and venular flow changes, they don't say why this is advantageous for either basic or clinical research studies of regional brain function. They also do not address the challenges in translating this approach to human studies where fields strengths are lower, the cortical surface is further from the RF coil, and subjects are not paralyzed to minimize motion. Accordingly, there is zero basis in the manuscript for the conclusion that the method has translational potential for identifying vascular dementia, or what its potential advantages are as compared to existing methods. Indeed, based on Figure 3 it would appear that using PC-MRI to identify voxels of interest for BOLD contrast changes might provide superior data quality to dynamic PC-MRI. Thus, the potential to experimentally address a previously inaccessible biological question is not demonstrated in this work.

Reviewer #3: In this work the authors have implemented and evaluated a magnetic resoance imaging (MRI) method to measure cerebral blood flow (CBF) with singel vessel resolution in the rat brain, based on the phase contrast (PC) technique. This method complements and is a valuable addition to the established BOLD and CBV techniques to assess brain function by MRI. The authors have previously pioneered single vessel fMRI, and now combine their recent methods with PC-MRI. The novel method is a valuable asset for functional brain imaging and may help to further understand mechanisms of neurovascular coupling and brain function in general.

The work is carefully conducted, the manuscript well written, and of great relevance and interest to the fMRI neuroiamging community. However, for a wider readership, the paper may be too detailed.

While I have no major points that would have to be addressed before the paper can be published, I think a few clarifications concerning reproducibility and possible applications are necessary:

1. The measurements were conducted at 14.1 tesla, a magentic field strength that is not available in most labs. Will this method also be feasible atthe more common 7 tesla or 9.4 tesla?

2. Which pulse program/MR sequence did the authors use; was it the sequence provided by the vendor Bruker?

3. For replication of these experiments, details such as the exact sequence and gradient timing (and ramps) are important. Is it possible to share these details perhaps even the source code of the method (in a supplement)?

4. Are the data presented in Fig. 2D values from single voxels or were they averaged (over how many voxels)? Figure S3 creates the impression that single vessels spread over five to nine voxels.

5. What is the minimum vessel diameter that can be analysed with the presented method? How would this be different at other field strengths or in humans? What was the average vessel diameter in this study?

6. In line 210 the author speculate on a translational potential. How can his method be translated? Is single vessel-CBF feasible in humans? What field strength would be required?

Typos:

l97: "white matter"

l275: missing "were"

Legend to Fig. 3: check for use of direct articles.

---

## [Decision Letter · Decision Letter 2]

8 Jul 2021

Dear Dr Yu,

Thank you for submitting your revised Methods and Resources entitled "Single-vessel cerebral blood velocity fMRI based on phase-contrast imaging" for publication in PLOS Biology. I have now obtained advice from the original reviewers and have discussed their comments with the Academic Editor. 

The reviews are appended below. As you will see, reviewers 1 and 3 are completely satisfied by the revision and think the manuscript has been substantially improved. Reviewer 2 also has commented that the revision has addressed many of his/her concerns, but notes that the manuscript should be thoroughly edited for grammar and for clarity.

Based on the reviews, we will probably accept this manuscript for publication, provided you satisfactorily address the remaining points raise by the reviewer 2 and carefully edit the manuscript for English syntax. IMPORTANT: Please also make sure to address the following data and other policy-related requests:

1) Ethics request: In the methods section of your manuscript, please also include an approval number for the animal care and use protocol approved by the Regierungspraesidium, Tübingen Referat 35, Veterinärwesen, Leiter Dr. Maas.

2) Data request: We note that your data availability statement states “Data are available from the Max Planck Institutional Data Access / Ethics Committee for researchers who meet the criteria for access to confidential data”. You may be aware of PLOS' Data Availability Policy (https://journals.plos.org/plosbiology/s/data-availability), which requires that all data be made available without restriction. In order to be compliant with our data availability policy, we will need you to provide the data underlying each figure, as either a supplementary data file or as a deposition in a publicly available repository. Please find the specific details of this request, in the ‘DATA POLICY’ section below my signature. **When addressing this request, you will also need to update your data availability statement to convey where the underlying data can be found. Please also add a sentence to each relevant figure legend (including supplemental) referencing the supplemental file or repository where the underlying data can be found - for example you can add the sentence "The data underlying this figure can be found in S1_data". 

3) Scale bars: Please provide scale bars for all figures in which images are shown. 

4) Title: Having discussed the title of your manuscript with my colleagues, we wonder if it might be edited slightly to improve clarity. If you agree, we might suggest that you change it to something like "Assessment of single-vessel cerebral blood velocity by phase contrast fMRI". 

We expect to receive your revised manuscript within two weeks. 

*Published Peer Review History*

*Early Version*

Sincerely,

Lucas Smith, Ph.D.,

Associate Editor,

lsmith@plos.org,

PLOS Biology

ETHICS STATEMENT:

-- Please include an approval number of the animal care and use protocol/permit/project license approved by the Regierungspraesidium, Tübingen Referat 35, Veterinärwesen, Leiter Dr. Maas.

DATA POLICY:

Figure 1C; Figure 2D; Figure 3 B-F; Figure S2A-B; Figure S4C-D; Figure S5B;

Reviewer #1, Ravi L. Rungta: The authors have addressed my comments and I congratulate them on a nice technical study. I support the paper for publication.

Reviewer #2: The revised manuscript has addressed many of my concerns, and is also supported by new data. The findings of this study remain interesting. However, some of the modifications to the manuscript still need to be revised for clarity. The manuscript also needs careful editing for English syntax.

1. Line 221: "focal" RF coil should be "local" RF coil

2. Line 223: "diameter" (or radius) is missing from this sentence

3. Line 230-234: The paragraph beginning "In contrast to the T2* weighted resting-state" leads the reader to expect something about differing contrast mechanisms, but the sentence winds up being about frequency (CBFv signal fluctuation … has seldom been directly measured …" Why is this statement in the middle of a paragraph about MRI-based CMRO2 measurements? Is there really a need to mention that their novel approach has been used less often than the most widely used approach?

4. Lines 234-237: Then this same paragraph concludes with a sentence suggesting that this approach can be used to study glymphatic function without explaining how. Paragraphs should have clear themes.

5. Line 238: The next paragraph switches to hardware limitations of phantom studies, but without clearly indicating that except through saying "phantom capillary tubing". Paragraphs should have clear themes.

6. Lines 251-253: This sentence seems to be addressing partial volume effects, but doesn't ever say that directly.

7. Lines 258-267: This paragraph appears to be an effort to respond to the need for better description of the translational (to humans) potential of this method, but the first two sentences discuss animal studies at 9.4T, the third sentence discusses a different type of human functional PC study. The fourth sentence mentions "the pulsation" as a potential confound without explaining what that is, and the last sentence mentions some technical challenges without explaining why they are important.

8. Also, in response to Reviewer 1, the authors have replaced "CBF-fMRI" to "CBFv" which is at best minimally better that CBF-fMRI in differentiating the PC MRI measure (velocity) from CBF.

Reviewer #3, Cornelius Faber: The authors have satisfactorily addressed all points I had raised in my original review. The revised manuscript has improved substantially. I have no further comments.

---

## [Editor Report · Decision Letter 3]

28 Jul 2021

Dear Dr Yu,

On behalf of my colleagues and the Academic Editor, Aniruddha Das, I am pleased to say that we can in principle offer to publish your Methods and Resources "Assessment of single-vessel cerebral blood velocity by phase contrast fMRI" in PLOS Biology, provided you address any remaining formatting and reporting issues. These will be detailed in an email that will follow this letter and that you will usually receive within 2-3 business days, during which time no action is required from you. Please note that we will not be able to formally accept your manuscript and schedule it for publication until you have made the required changes.

Your manuscript has been evaluated by the editorial team and the Academic Editor, and overall we are largely satisfied by your response to the reviewers and our previous editorial requests. However, we do have two additional and minor editorial requests which we need you to address along with the formatting and reporting requests which that are soon to come. 

IMPORTANT: Please address the following points:

1) Thank you for providing, as supplementary files, the data underling your figures. As you make the final edits to your manuscript, we additionally ask that you add a sentence to each figure legend (including supplemental) referencing the supplemental file where the underlying data can be found. For example you can add the sentence "The data underlying this figure can be found in supplementary file ‘Main_Figure_data’".

2) We think the sentence starting on line 261 ("Previously, we detected low-frequency fluctuation of single-vessel resting state BOLD fMRI signals (TR = 1 s) from individual sulcus veins and arteries in the occipital lobe of the human brain [38], bearing the interference of pulsation and other motion effects.) requires editing for clarity. In particular, the meaning of 'bearing the interference of pulsation..." is unclear. Do you mean barring? resulting from? Additionally, we think the segue to the next sentence is similarly unclear: "Different from ..." and would be worth clarifying.

PRESS

Sincerely, 

Lucas Smith, Ph.D. 

Senior Editor 

PLOS Biology

lsmith@plos.org